# Deeply Supervised UNet for Semantic Segmentation to Assist Dermatopathological Assessment of Basal Cell Carcinoma

**DOI:** 10.3390/jimaging7040071

**Published:** 2021-04-13

**Authors:** Jean Le’Clerc Arrastia, Nick Heilenkötter, Daniel Otero Baguer, Lena Hauberg-Lotte, Tobias Boskamp, Sonja Hetzer, Nicole Duschner, Jörg Schaller, Peter Maass

**Affiliations:** 1Center for Industrial Mathematics, University of Bremen, 28359 Bremen, Germany; nick7@uni-bremen.de (N.H.); otero@uni-bremen.de (D.O.B.); hauberg@math.uni-bremen.de (L.H.-L.); pmaass@uni-bremen.de (P.M.); 2SCiLS, Bruker Daltonik, 28359 Bremen, Germany; tobias.boskamp@bruker.com; 3Dermatopathologie Duisburg Essen, 45329 Essen, Germany; sonja.hetzer@web.de (S.H.); nicole.duschner@gmail.com (N.D.); dermatohistologie@googlemail.com (J.S.)

**Keywords:** digital pathology, dermatopathology, whole slide image, basal cell carcinoma, skin cancer, deep learning, UNet

## Abstract

Accurate and fast assessment of resection margins is an essential part of a dermatopathologist’s clinical routine. In this work, we successfully develop a deep learning method to assist the dermatopathologists by marking critical regions that have a high probability of exhibiting pathological features in whole slide images (WSI). We focus on detecting basal cell carcinoma (BCC) through semantic segmentation using several models based on the UNet architecture. The study includes 650 WSI with 3443 tissue sections in total. Two clinical dermatopathologists annotated the data, marking tumor tissues’ exact location on 100 WSI. The rest of the data, with ground-truth sectionwise labels, are used to further validate and test the models. We analyze two different encoders for the first part of the UNet network and two additional training strategies: (a) deep supervision, (b) linear combination of decoder outputs, and obtain some interpretations about what the network’s decoder does in each case. The best model achieves over 96%, accuracy, sensitivity, and specificity on the Test set.

## 1. Introduction

Deep learning has shown great potential to address several problems in understanding, reconstructing, and reasoning about images. In particular, convolutional neural network approaches have been actively used for classification and segmentation tasks in a wide field of applications [1,2,3], ranging from robot vision and understanding to the support of critical medical tasks [4,5,6,7]. The nearly human-expert performance achieved in some medical imaging applications [8,9] has come to show the capabilities and potential of these algorithms. It also enabled the extraction of previously hidden information from routine histology images and a new generation of biomarkers [10]. In this work, we aim to develop a deep learning method that can support dermatopathologists in providing fast, reliable, and reproducible decisions for the assessment of basal cell carcinoma (BCC) resection margins.

BCC is the most common malignant skin cancer with an increasing incidence of up to 10% a year [11]. It can be locally destructive and is an essential source of morbidity for patients, mainly when located on the face. Thus, it must be adequately treated, despite its slow growth [12]. Although BCC can be effectively managed through surgical excision, determining the most suitable surgical margins is often not trivial. Complete removal of the pathological tissue is the key to a successful surgical treatment. Initially, the tumor is removed with a safety margin of surrounding tissue, and it is sent to a laboratory for analysis. If remaining tumor parts are detected in the margins, further surgery may be performed. For BCC, approximately 10% of the tumors recur after the usual removal surgeries [13]. The resection margins’ microscopic control can reduce the recurrence rate to 1% [14,15].

As a part of the laboratory routine, each tissue sample is cut into several slices. The dermatopathologist can only be sure that the surgery was successful if no tumor is present in the ones close to the margins. This task consists of several time-consuming and error-prone manual processes, including examination under the microscope, which is still the most common practice. The adoption of digital pathology offers several benefits in terms of time savings and performance [16], and also the opportunity to use a computer-aided diagnostic system, which can potentially improve diagnostic accuracy and efficiency, as demonstrated in recent studies [17].

Besides the success of deep learning methods in various histology imaging tasks [18,19,20,21], to the best of our knowledge, there are only a few works [22,23,24] on tumor recognition in dermatopathological histology images. We argue that the automated diagnostic of microscopic images of the human skin can be incredibly challenging due to the large variety of relevant data. Dermatological features like the structure of the extracted tissue depend on several aspects, e.g., the body part or the patient’s skin type. Therefore, the pursued deep learning model has to be more robust toward such changes than models that segment histology images of inner organs. An extensive database, such as the one we use for this study, is essential for reliable digital dermatopathology solutions. While a previous study [22] obtained promising results in diagnosing a specific BCC subtype, we focus on the automatic detection of BCC in general, including several subtypes such as sclerodermiform, nodular, and superficial.

The first and most crucial step we do is to automatically highlight where the tumors are located with the highest probability. That also allows the dermatopathologist to interpret the computer-based decisions better. To this end, we obtain a deep-learning-based semantic segmentation of a WSI into two classes: Tumor and Normal, i.e., each pixel of tissue in the image is assigned to one of these labels. Following, we decide for each section whether it contains tumors or not based on the segmentation and account for possible model errors. Most recent semantic image segmentation methods are based on fully convolutional networks [25]. In this work, we use UNet [4] as the base architecture, which has been successfully applied in a large variety of medical image analysis applications [5,26,27]. In another recent work, UNet was also used to segment the epidermis from the skin slice [28].

The structure of the manuscript is organized as follows. In Section 2, we describe the methods used for the data collection and some statistics of the different data parts used in the study. There, we also describe the different model architectures and training strategies that we compare. Section 3 contains the final results on the Test part of the data. Section 4 contains some discussion and analysis of the results, as well as interesting interpretations of the proposed training strategies. Finally, Section 5 contains the conclusions of the work, and in Section 6, we describe some limitations and possible future directions.

## 2. Methods

### 2.1. Data Collection and Data Parts

The tissue samples belong to primary excisions, biopsies, and marginal excisions. They were processed in the “Dermatopathologie Duisburg Essen” laboratory using standard protocols. In the selection of the samples for the study, no distinction was made regarding the type of excision it was. After fixation in formalin and the usual paraffin embedding, the samples were cut into slices of ≈3 μm thickness, and standard hematoxylin and eosin (H&E) staining was applied. The glass slides were digitized using a “Hamamatsu NanoZoomer S360” scanner with a 20× lens. In total, this study includes 650 WSI annotated in different manners and used for training, validation, and testing. Two clinical dermatopathologists annotated 100 WSI, which were included in the Train and Validation I parts of the data (see Table 1). The annotations contain different interest regions of the tissue: *Tumornest* indicates where tumor islands are, *Stroma* highlights supporting tissue around the tumor islands, and *Normal* contains distractors such as hair follicles, actinic keratosis, or cysts. However, only the *Tumornest* annotations were done exhaustively, whereas the others were annotated only in some cases. For that reason, we only use *Tumornest* annotations for training and leave the other types for data balancing. Some annotation examples are shown in Figure 1.

Due to the huge number of WSI, it was not feasible to create detailed annotations for all the slides. Two parts of the data, namely, Validation II and Test, only contain sectionwise labels. That means that for each section of tissue in the WSI, there is a label that indicates whether there is some tumor inside or not (see Figure 2).

### 2.2. Blind Study and Web Application

The Test data were provided by the dermatopathologists in an advanced stage of the study to evaluate the models in the form of a blind study. That means it was not the result of shuffling and random splitting the available data as is typically done. At the moment of training, the sectionwise annotations for this part of the data did not exist yet. We first generated bounding boxes for the tissue sections and assigned the labels Normal or Tumor based on one of our models. The dermatopathologists then corrected these labels by just changing those that were wrong. That allowed us to obtain the sectionwise annotations of the Test data in a time-efficient manner. The process was conducted using our experimental web application called Digipath Viewer (https://digipath-viewer.math.uni-bremen.de/), which was created for reviewing and visualizing WSI, annotations, and model predictions. This application was also used to obtain some of the detailed annotations for the Training and Validation I data. It was built on ReactJS and Python using Openslide [29].

### 2.3. Model Architecture

The neural network designs we use in this work follow the UNet architecture [4], a fully convolutional encoder–decoder network with skip connections between the encoder blocks and their symmetric decoder blocks. We use two different encoders and a standard decoder, similar to the original one [4] with only minor changes. The input to the network is a patch $x\in R^{512\times 512\times 3}$, and the output is a segmentation map $\Phi \left(x\right)\in {\left[0,1\right]}^{512\times 512\times 2}$. The segmentation map contains two matrices of size 512×512, which correspond to the predicted pixelwise probabilities for each of the two classes: Tumor and Normal. Since we use the Softmax activation to compute the segmentation map, this can be seen as only one matrix of size 512×512 with the Tumor probabilities.

#### 2.3.1. Encoder

The first encoder follows nearly the same design as the original one [4] with 5 blocks, which successively downsample the spatial resolution to increasingly catch higher-level features. Each of the blocks doubles the number of feature channels and halves the spatial resolution. It uses two 3×3 convolutions, where the first one uses a stride of 2. The second encoder is exactly the convolutional backbone of a ResNet34 [30], as can be observed in Figure 3, and also contains 5 blocks.

Both encoders contain an initial block with a 7×7 convolution with 64 filters and a stride of 2, which decreases the input resolution by half. That allowed us to effectively enlarge the patches’ size (512×512) to incorporate a larger context without a significant increase in computation and memory consumption.

#### 2.3.2. Decoder

The decoder contains an expanding path that seeks to build a segmentation map from the encoded features (see Figure 3). It has the same number of blocks as the encoder. Each decoding block duplicates the spatial resolution while halving the number of feature channels. It performs a bilinear upsampling and concatenates the result with the output of its symmetric block in the encoder (skip connection). Next, it applies two 3×3 convolutions with the same number of filters. Unlike the original decoder proposed in [4], it has one extra block that does not obtain any skip connection. The output of the last decoder block is passed to a 1×1 convolution that computes the final score map.

Additionally, at each block of the decoder, we added an extra 1×1 convolution, i.e., a linear combination of the block’s final feature maps, to produce an intermediate score map. We used these maps in two settings: (a) deep supervision and (b) to merge them through a linear combination to produce the final output. Except for the 1×1 convolutions, we always use a batch normalization layer [31] and a ReLU activation function right after each convolution in the whole network. Furthermore, we always use the corresponding padding to keep the same spatial size unchanged by the convolution’s kernel.

### 2.4. Model Training

For training the models, we used the Train set, which contains 85 annotated WSI. We extracted small patches of size 512×512 at 10× level of magnification, on which we performed the semantic segmentation. In almost all slides, the tumor-free tissue is dominant; therefore, it was necessary to balance the training data to avoid biases and improve the performance. We did this based on the percentage of pixels belonging to the 3 types of annotations: *Tumornest* (*T*), *Stroma* (*S*), and *Normal* (*N*). The resampling was done as depicted in Table 2. That means that some of the patches were used several times during a training epoch. That is the case for patches that contain tumor, are close to tumor areas (*Stroma*), or contain distractors (*Normal*). Additionally, we did extra oversampling for patches with high tumor density.

During training, all patches were extensively augmented using: random rotations, scaling, smoothing, color variations, and elastic deformations to increase the variety of the data effectively. Additionally, we used Focal-Loss with γ=2.0 [32], which has a similar effect to downweighting the easy examples, to make their contribution to the total loss smaller. We did not include any resampling or augmentation for the validation data set.

We trained the models using a maximum of 40 epochs and a batch-size of 64. Optimization was performed with the Adam method [33], a learning rate of $5\times 10^{-4}$, and a scheduler to multiply it by 0.8 every 5 epochs. All computations were done on a server running Ubuntu 16.04 and equipped with a 24-core processor, 1.5 TB RAM, and 4 NVIDIA GeForce GTX 1080 Ti (11 GB GPU memory). We implemented our models using the Pytorch [34] deep learning library.

#### 2.4.1. Deep Supervision

The deep supervision strategy consists of forcing the decoder blocks’ outputs to yield a meaningful segmentation map. We compare each of the decoder blocks’ output with the corresponding downsampled version of the target segmentation map and add the discrepancy to the total loss. This technique was originally introduced in [35] for obtaining transparency and robustness of the features extracted in the middle of the network and helping address the vanishing gradient problem. In this case, it allows gradient information to flow back directly from the loss to every block of the decoder. Some recent works [36,37] used a similar idea for training a UNet.

Let $x\in R^{512\times 512\times 3}$ be an input patch and its corresponding target segmentation map $y\in {\left\{0,1\right\}}^{512\times 512\times 2}$. We define $\psi_{\ell }\left(x\right)$ as the output of the *ℓ*-block of the decoder (after Softmax) (see Figure 3). The contribution to the loss function from this single data point (x,y) is then defined as
(1)$$loss\left(x,\;y\right)=\sum_{\ell =0}^{k-1}fl\left(\psi_{\ell }\left(x\right),\;\Pi_{\ell }\left(y\right)\right),$$
where k=5 is the number of decoder blocks, $\Pi_{\ell }:\;{\left\{0,1\right\}}^{512\times 512\times 2}\to {\left\{0,1\right\}}^{32\;\cdot \;2^{\ell }\times 32\;\cdot \;2^{\ell }\times 2}$ is a downsampling operator, and fl is the Focal-Loss. We also tried using different weights for each of the scales, but we did not observe any improvement. This strategy does not involve any change on the previously described architecture; it only needs the 1×1 convolution and the Softmax at the end of each block of the decoder. The final output is given by $\Phi \left(x\right)=\psi_{k-1}\left(x\right)$.

#### 2.4.2. Linear Merge

The second strategy merges the decoder outputs through a linear combination. We add a linear layer with weights $w\in R^{k}$ to the architecture that computes the final output, i.e.,
(2)$$\Phi \left(x\right)=\mathrm{Softmax}\left(\sum_{\ell =0}^{k-1}w_{\ell }\;\cdot \;\Gamma_{\ell }\left({\hat{\psi }}_{\ell }\left(x\right)\right)\right),$$
where ${\hat{\psi }}_{\ell }\left(x\right)$ is the score map computed at the *ℓ*-block of the decoder, i.e., the same as $\psi_{\ell }\left(x\right)$ but without the Softmax activation, and $\Gamma_{l}:\;R^{32\;\cdot \;2^{\ell }\times 32\;\cdot \;2^{\ell }\times 2}\to R^{512\times 512\times 2}$ is a bilinear upsampling operator. In this case, the Softmax is only applied after the linear combination. The weights *w* are trained together with the other parameters of the model. Note that the standard model without this strategy is a special case, since the model could learn to assign $w_{k-1}=1.0$ and $w_{\ell }=0$ for 0≤ℓ<k.

### 2.5. Sectionwise Classification

The final classification task for each section of the WSI (see Figure 2) is to decide whether it contains tumor or not. First of all, we use the trained model to generate a heatmap by combining several patches’ predictions and highlighting the tissue’s parts with the highest probabilities to contain tumor. We extract the patches so that they cover the whole tissue area and have at least 256 pixels (50%) overlapping at every border. Following, we apply a prediction threshold to obtain a binary mask and find connected regions. To account for some possible model errors and reduce the false positive rate, we filter out regions below a certain area threshold. If any predicted tumor area is left in the section, it is classified as Tumor; otherwise, it is classified as Normal. The prediction and area thresholds are selected for each model independently as part of the model selection.

### 2.6. Model Selection

We trained our implementation of the original UNet and three other variants using a ResNet34 encoder. The first alternative does not employ any decoder strategy (ResNet34-UNet), the second one uses the deep supervision strategy (ResNet34-UNet + DS), and the third one uses the linear combination of the decoder outputs (ResNet34-UNet + Linear). During training, we evaluated the models on the Validation I part of the data. We used the Intersection over Union (IoU) metric to select the models from the best five training epochs for each setting. Afterward, we evaluated these models on the sectionwise classification task in the Validation I and Validation II parts of the data to select the best model together with the prediction and area thresholds. To this end, we used a grid search and the $F_{\beta }$ score
(3)$$F_{\beta }=\frac{(1+\beta^{2})\;\cdot \;\mathrm{precision}\;\cdot \;\mathrm{recall}}{\beta^{2}\;\cdot \;\mathrm{precision}+\mathrm{recall}},$$
with β=1.5, to give higher importance to the recall/sensitivity.

The performance of the best models and the selected thresholds are shown in Table 3. Additionally, Figure 4 presents the outputs of the models and the resulting masks after applying the corresponding thresholds for some patches from the Validation I data.

## 3. Results

Finally, we evaluated the selected models from the validation phase on the Test data. The results are depicted in Table 4 and Figure 5 shows the number of wrongly classified sections by each model. The ResNet34-UNet + DS achieved the best results, obtaining more than 96% overall accuracy, sensitivity, and specificity. It wrongly classified only 71 (30 FP and 41 FN) out of 1962 sections. Figure 6 shows some correctly classified sections for different BCC subtypes and the corresponding heatmaps generated with this model. In most false negative cases, the model detects the tumor but without enough confidence. Due to the selected prediction and area thresholds, these sections are then classified as Normal. On the other hand, false positives are often due to a hair follicle or other skin structure that is identified as a tumor (see, for example, Figure 7).

The ResNet34-UNet + DS was the second-best in the model selection phase. We argue that the slightly different results on the test set are due to the fact that the validation data is biased. Initially, the Validation II part was larger, but we selected challenging slides, based on the dermatopathologists’ feedback, which were then fully annotated and included in the Training part. All in all, the classification error was reduced from 8.4% (baseline UNet) to 3.6%. The baseline UNet, which uses the standard encoder, has excellent sensitivity but quite a low specificity. In contrast, the models with the ResNet34 encoder are more balanced and exhibit better performance.

## 4. Discussion and Interpretability

We now analyze in more detail the models trained with the two additional strategies, namely, ResNet34-UNet + DS and ResNet34-UNet + Linear. In Figure 8, we show the decoder outputs for some patches from the Validation I data using these models.

In the deep supervision case, one can observe that, indeed, it is possible to guide the UNet to produce a meaningful segmentation already from the first block ($\psi_{0}$) of the decoder, see Figure 8. The difference between $\psi_{0}\left(x\right)$ and $\psi_{4}\left(x\right)$ is only the resolution and the number of details on the borders. We observed that already in $\psi_{0}\left(x\right)$, the model knows where the tumor is if there is any. Therefore, we separately evaluated the performance of $\psi_{0}$ and the other decoder blocks on the final classification task, using the same prediction and area threshold selected for the original model. The results are nearly the same as when using the whole model, see Table 5; in the case of $\psi_{0}$, the difference is only 5 sections, whereas in $\psi_{1}$ it is only 1. Those sections were wrongly classified because the predicted tumor area was right at the limit.

Moreover, performing inference using the encoder followed by only the decoder’s first block $\psi_{0}$ is much faster and boosts the heatmap generation process’s speed. The original model takes approximately 30 s per WSI (several sections) on average on a NVIDIA GeForce GTX 1080 Ti. In contrast, the reduced one needs approximately only 20 s, which represents a reduction of 33% of the time.

On the other hand, the setting that uses a linear merge of the decoder blocks’ outputs has a fascinating behavior. In this case, the model has more freedom since we do not guide any decoder block to output a meaningful segmentation. We only include the final linear combination in the loss function. In Figure 8, one can observe that in $\psi_{0}\left(x\right)$, the model identifies a rough approximation of the tumor’s location, which is much larger than the final result. Following, in the subsequent blocks, it creates more details. The learned weights for each block are w=[0.1590,−0.1645,0.1603,−0.2963,0.0036]. Even though the observed behavior makes sense and offers some interpretability, it seems that, at least for the final classification task, it does not bring any advantage. As we have seen, only one block of the decoder seems to be good enough. The linear merge strategy might be more effective if the aim is to obtain exact borders.

## 5. Conclusions

In this work, we used a UNet architecture with two different encoders and several training strategies for performing automatic detection of BCC on skin histology images. Training the network with deep supervision was the decisive factor for improving the final performance. After trained with this strategy, each decoder’s block focused on obtaining a segmentation with more details than the previous one but did not add or remove any tumor content. We found out that the decoder’s first block is enough for obtaining nearly the best results on the classification task. That implies a substantial speed improvement on the forward pass of the network and, therefore, on the heatmap generation process (33% time reduction). We performed the final evaluation on a rather large Test data set compared to the Training part of the data. Still, the best model obtained a 96.4% accuracy and similar sensitivity and specificity on the sectionwise classification task. These results are promising and show the potential of deep learning methods to assist dermapatopathological assessment of BCC.

## 6. Limitations and Outlook

One of the main limitations of the study is that all the tissue samples were processed in the same laboratory. Even though they presented different staining and appearance, as can be observed in Figure 6, it is not guaranteed that the model will perform well on data collected in a different laboratory.

A limitation of the model is that it often detects a large area of a tumor but with low confidence. Thus, after applying the threshold, the remaining area is not enough to be classified as a tumor. An example of this behavior is shown in Figure 7d), where the section is wrongly classified as Normal. A slightly different approach would be to extract additional features [19] from the heatmaps and train a more complex classifier. Another alternative would be to include, together with the classification result, a confidence level. As suggested in recent works [24], it would certainly give more information to the dermatopathologists and filter out predictions that are unlikely to be correct.

## Figures and Tables

**Figure 1 jimaging-07-00071-f001:**
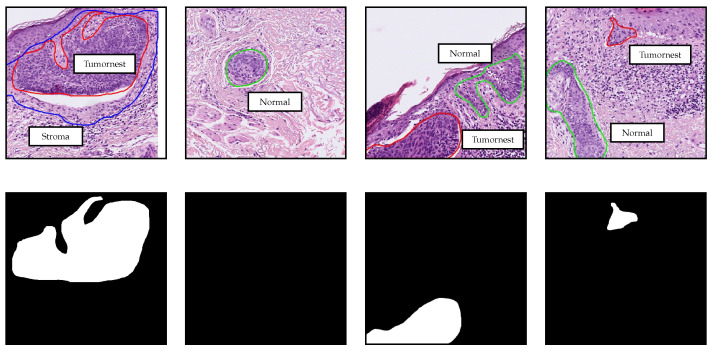
Detailed annotations used for training, validation, and data balancing. Regions delimited by red, blue, and green contours from the first row correspond to *Tumornest*, *Stroma* and *Normal* annotations respectively. The second row contains the masks for the *Tumornest* annotations in the first row. All patches have 512×512 pixels and are extracted at 10× magnification.

**Figure 2 jimaging-07-00071-f002:**
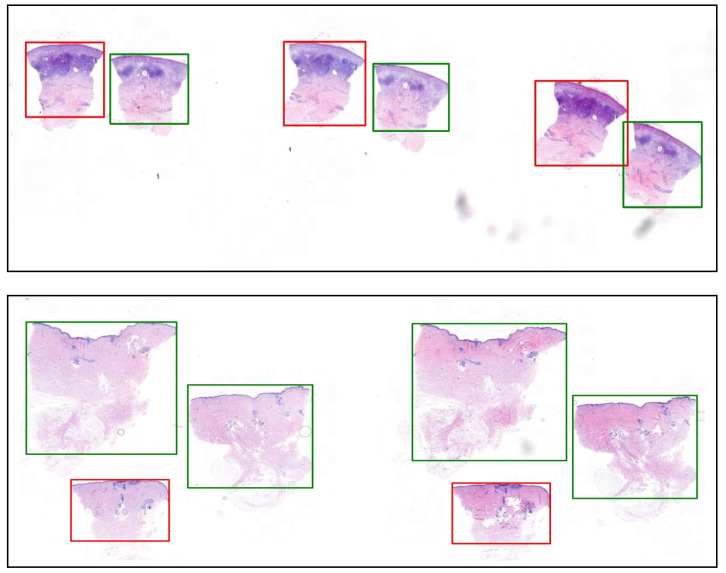
Example ground-truth labels (Tumor in red and Normal in green) for two whole slide images (WSI) from the Test data.

**Figure 3 jimaging-07-00071-f003:**
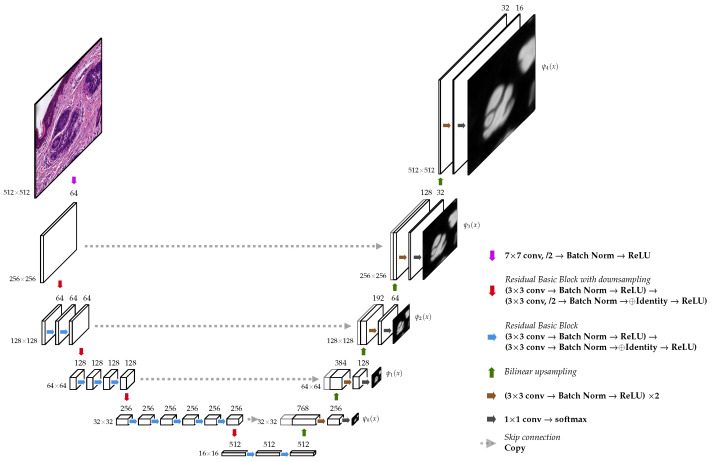
UNet architecture with a ResNet-34 encoder. The output of the additional 1×1 convolution after Softmax is shown next to each decoder block.

**Figure 4 jimaging-07-00071-f004:**
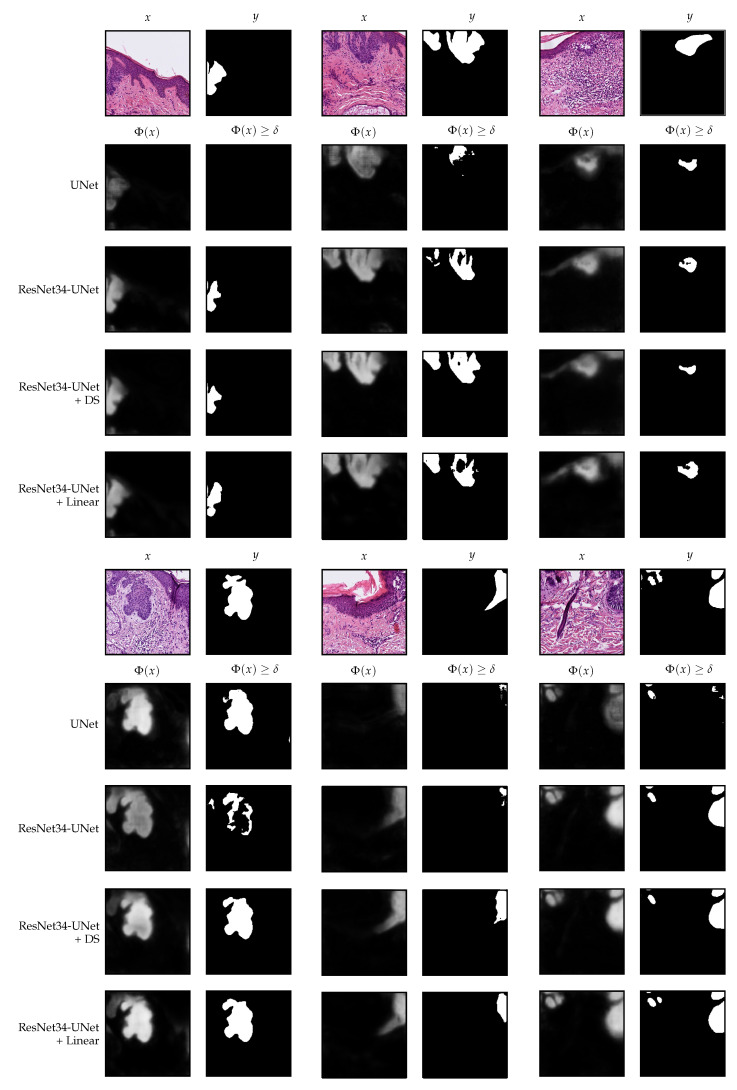
Segmentation examples using the analyzed settings on patches from the Validation I part of the data. All patches have 512×512 pixels and are extracted at 10× magnification.

**Figure 5 jimaging-07-00071-f005:**
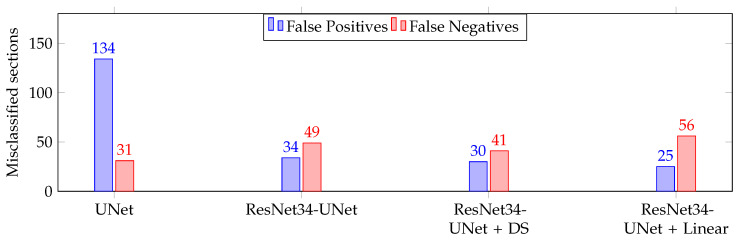
Number of wrongly classified sections on the *Test* part of the data. There are 1962 sections in total.

**Figure 6 jimaging-07-00071-f006:**
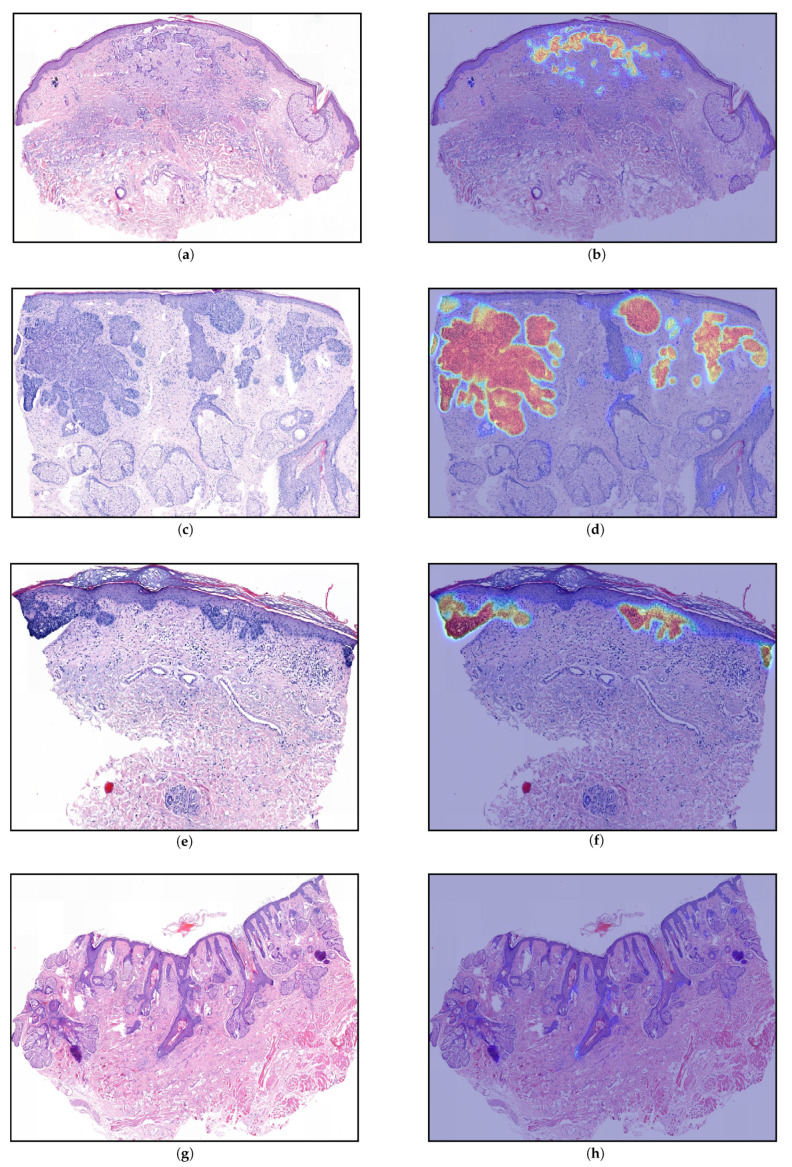
Generated heatmaps (ResNet34-UNet + DS) for sections from the Test part of the data. The images show a variety of basal cell carcinoma (BCC) subtypes that were part of the data set: (**a**,**b**) sclerodermiform BCC, (**c**,**d**) nodular BCC, (**e**,**f**) superficial BCC, (**g**,**h**) no tumor. As heatmap (**b**) suggests, the exact segmentation of sclerodermiform BCC can be quite challenging. In all cases, the heatmaps were qualitatively evaluated by the dermatopathologist and all the detected areas (orange-red) correspond to tumors, whereas there is no tumor that was not detected. The largest connected areas above the threshold (0.60) in (**c**,**d**,**f**) have 63,744 $\mu m^{2}$, 509,600 $\mu m^{2}$, and 37,632 $\mu m^{2}$ respectively.

**Figure 7 jimaging-07-00071-f007:**
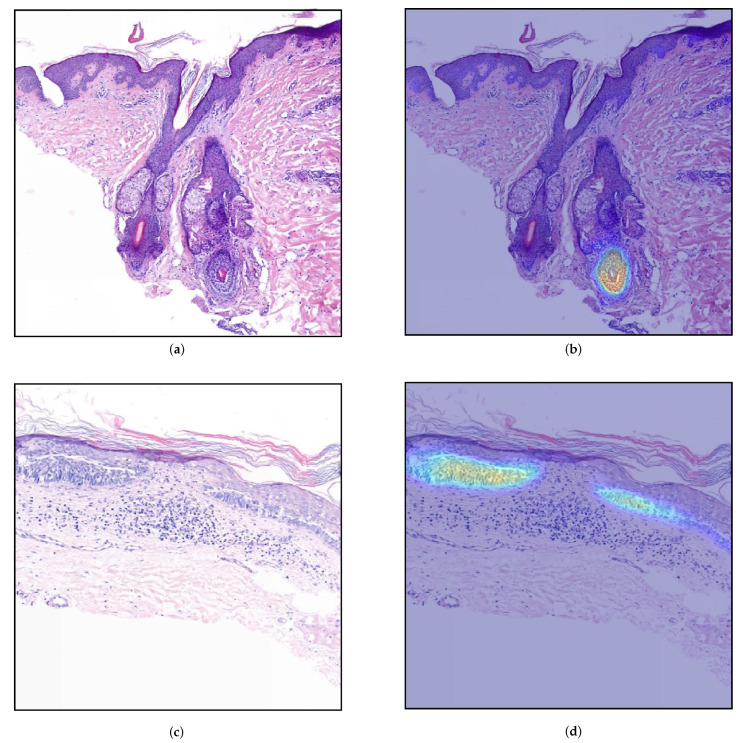
Generated heatmaps (ResNet34-UNet + DS) for sections from the Test part of the data that were wrongly classified: (**a**,**b**) false positive, the model wrongly identified a hair-follicle as a tumor (10,656 $\mu m^{2}$); (**c**,**d**) false negative, the model detected the BCC but not with enough confidence, i.e., the largest connected area above the threshold (0.60) was too small ( 544 μ$m^{2}$).

**Figure 8 jimaging-07-00071-f008:**
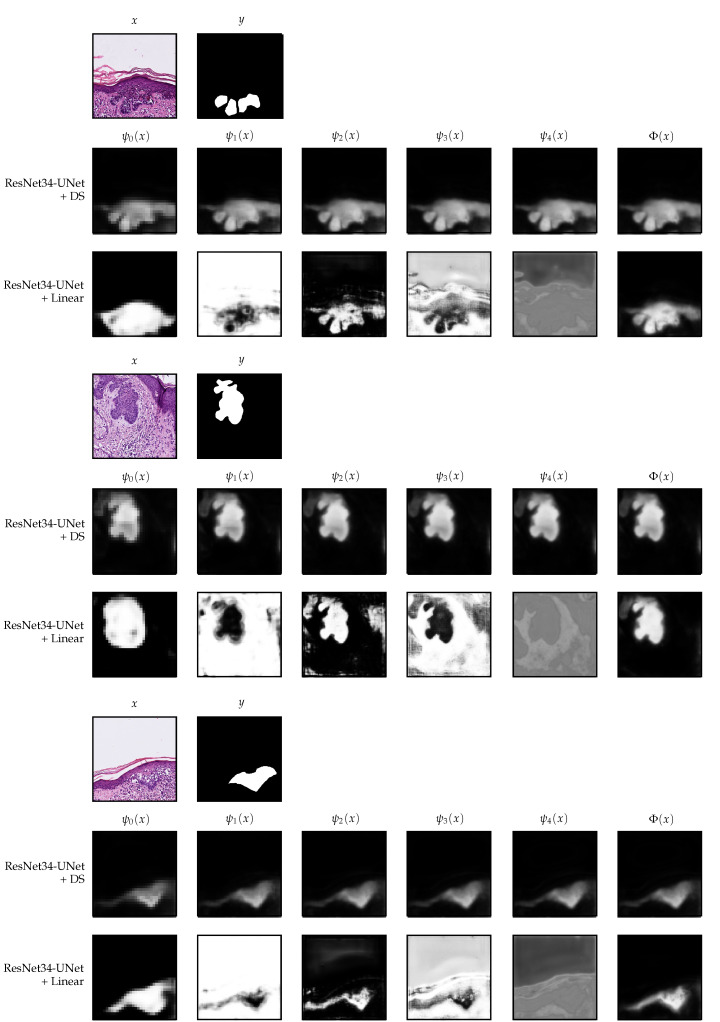
Decoder outputs for each block of the decoder in the deep supervision and linear merge settings. The patches belong to the Validation I part of the data. For the linear merge strategy, the segmentation maps shown in the figure are after applying the Softmax operation, which we do in this case only for visualization purposes. All patches have 512×512 pixels and are extracted at 10× magnification.

**Table 1 jimaging-07-00071-t001:** Distribution of the slides, sections, and type of annotations. All slides contain sectionwise annotations with Tumor or Normal labels.

Part	Slides	Detailed Annotations	Tumor Sections	Normal Sections
Training	85	✓	188	209
Validation I	15	✓	53	31
Validation II	229		392	608
Test	321		1119	843

**Table 2 jimaging-07-00071-t002:** Patches distribution and pixelwise unbalance before and after resampling the data. Each row indicates the number of patches in which the given annotation type covers the indicated percentage of pixels, e.g., the first one shows that the patches with less than 0.05% of pixels covered by *Tumornest* (*T*) annotations were not resampled. The pixel unbalance is computed as the ratio between the amounts of tumor-free and tumor pixels.

	Before	After
T<0.05%	175,771	175,771
T≥0.05%	9537	30,000
T≥10%	5528	10,000
S≥0.05%	9096	20,000
N≥0.05%	9458	20,000
Total patches	209,390	255,771
Pixel unbalance	78.48	16.81

**Table 3 jimaging-07-00071-t003:** Results on the sectionwise classification task for all sections from the Validation I and Validation II parts of the data. The values correspond to the best models and selected thresholds. The highest accuracy and $F_{\beta }$ score are highlighted in bold font.

Setting	Prediction Threshold	Tumor-Area Threshold ($m^{2}$)	Accuracy	$F_{\beta }$
UNet	0.45	8960	0.985	0.989
ResNet34-UNet	0.60	3840	0.993	0.994
ResNet34-UNet + DS	0.60	5120	0.994	0.993
ResNet34-UNet + Linear	0.65	2560	**0.996**	**0.997**

**Table 4 jimaging-07-00071-t004:** Results on the sectionwise classification task for all sections from the Test part of the data. The highest value for each metric is highlighted in bold font.

Setting	Accuracy	Sensitivity	Specificity	$F_{\beta }$
UNet	0.916	**0.972**	0.842	0.945
ResNet34-UNet	0.958	0.956	0.960	0.960
ResNet34-UNet + DS	**0.964**	0.963	0.965	**0.966**
ResNet34-UNet + Linear	0.959	0.950	**0.970**	0.958

**Table 5 jimaging-07-00071-t005:** Results on the test data for the ResNet34-UNet + DS, where the heatmaps are generated using the output of each decoder block. The highest value for each metric is highlighted in bold font.

Block	Accuracy	Sensitivity	Specificity	$F_{\beta }$
$\psi_{0}$	0.9612	0.961	0.961	0.964
$\psi_{1}$	0.9633	0.963	0.963	0.966
$\psi_{2}$	0.9633	0.963	0.963	0.966
$\psi_{3}$	**0.964**	**0.963**	**0.965**	**0.966**
$\psi_{4}$	**0.964**	**0.963**	**0.965**	**0.966**

## Data Availability

The data presented in this study are available on request from the corresponding author.

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
