# Peer review of "Deeply Supervised UNet for Semantic Segmentation to Assist Dermatopathological Assessment of Basal Cell Carcinoma"

_2313-433X, 2021, doi:10.3390/jimaging7040071_

Round 1

Reviewer 1 Report

Dear Authors,

After thoroughly reviewing your manuscript, I can conclude that it could be interesting to the journal's readers. However, several minor to moderate improvements should be made.

Please address the attached comments.

Kind regards,

Reviewer

Reviewer 2 Report

The article by Le’Clerc Arrastia, use machine learning for a specific application of detecting basal cell carcinoma in the skin. This is an interesting study worthy to be published.

The article needs to be improved by addressing following issue:

The authors describe their process at the end of the article in the blind section. This should be moved to the methods section.

L26 and 71- probe? Do authors mean lesion?

L30-31- this type of argument is prevalent in pathology AI papers, unfortunately there is no real-world evidence for it, as there are very few studies on error rate, and even less on pathologist workload.

L67- the type of sample should be described, were these punch biopsies of original diagnosis, or resection of known bcc for evaluation of the margin? Are there any specific evaluation of margin sections for residual BCC?

L78- “look similar to tumor” should be deleted it is confusing and defy the concept of classification; it also suffers from primary bias of selection of specific entities.

In addition, the labelling software/tools should be described here not as a footnote at the end of the article.

Table 2- please expand description of what the percentages mean (0.05%)

L 140- underlying library should be described (pytorch?dl4j?). Hardware and OS should be described.

Figure 6- please add metrics for each case

Figure 7- please add more examples of false negative with metrics

Round 2

Reviewer 1 Report

Dear Authors,

I find the conducted revision sufficient and adequate. 

Good job!

Reviewer 2 Report

-